# Service reliability evaluation of highway tunnel based on digital image processing

**Chunquan Dai**[1]*, **Zhaochen Zhou**[2], **Huidi Zhang**[2], **Kun Jiang**[2], **Haisheng Li**[3], **Haiyang Yu**[3]

1 Shandong Civil Engineering Disaster Prevention and Mitigation Laboratory, Shandong University of Science and Technology, Qingdao, China, 2 School of Civil Engineering and Architecture, Shandong University of Science and Technology, Qingdao, China, 3 Research and Development Department, Ronghua Construction Group Limited, Qingdao, China

* dcqwin@sdust.edu.cn

**Data Availability Statement:** All data supporting the findings of this study are available within the paper and its Supporting Information files.

**Funding:** The authors declare that they have no known competing financial interests or personal

## Abstract

China is gradually transitioning from the "tunnel construction exploration era" to the "tunnel high-quality construction and operation era", and the maintenance demand of highway tunnels has increased sharply. Therefore, there is an urgent need for an evaluation method to evaluate the service reliability of highway tunnels, so as to provide reference for tunnel maintenance personnel to carry out maintenance work. Taking highway tunnels as the research object, this paper extracts three parameters, including length, maximum width and fractal dimension, from the binary image of highway tunnel lining cracks. The standard for dividing the length of the highway tunnel section is 500m as the tunnel section, and a section disease sample space including multiple highway tunnels is constructed. The EM clustering algorithm was used to determine the number of graded grades of disease, and the relative Euclidean distance was used as the evaluation index to divide the safety grade of the tunnel into five grades: normal, degraded, inferior, deteriorated and hazardous. The partial least squares method is used to establish the lining service reliability evaluation formula and verify the residual of each sample point in the sample space. The smaller the average value of the residual, the better fitting effect of the established evaluation formula. The service reliability evaluation method proposed in this paper is applied to engineering practice and compared with the expert scoring method and the national standard method, which proves that the evaluation method in this paper has the advantages of strong visibility, simple evaluation method, and is conducive to engineering practice.

## 1. Introduction

China has become the country with the longest road tunnel mileage, the largest scale and the fastest development in the world [1]. More and more highway tunnels have reached a certain operating life. Due to the influence of geological conditions, hydrological conditions, design, construction, operation and maintenance and other factors, the tunnel lining will suffer from different degrees of disease, among which tunnel cracks are particularly prominent [2]. If the cracks generated are ignored, it is likely to lead to further deterioration of the cracks,

relationships that could have appeared to influence the work reported in this paper.

**Competing interests:** The authors declare no conflict of interest.

threatening the healthy operation of the tunnel, and reducing the service life of the tunnel. Therefore, an evaluation method is urgently needed to evaluate the service reliability of the highway tunnel, so as to provide reference for the tunnel maintenance personnel to carry out maintenance work.

To evaluate the service reliability of highway tunnels, it is first necessary to extract the parameters of cracks. Scholars from different countries have proposed different methods. Shigeta et al. [3] of Yamaguchi University in Japan extracted two basic parameters of fracture width and length, and on this basis, introduced a new strike parameter; Xu Jieying et al. [4] extracted three parameters of fracture area, concavity and compactness; Chiu et al. [5] took mountain highway tunnels as an example, extracted the development of cracks in different directions within a certain time range as qualitative parameters, and took the crack development speed and crack length as quantitative parameters to clarify the relationship between slope deformation and tunnel service reliability; Wang Pingrang [6] established the safety grade evaluation factor of tunnel structure with the length and width of crack disease as the main index and the trend, distribution density and position of crack disease as the auxiliary index; Wu et al. [7] of Nagasaki University introduced the concept of fractal dimension into the extraction of tunnel parameters, and proved that fractal dimension can describe the direction and complexity of tunnel cracks to a certain extent. Li Qingtong et al. [8] took the binary image of the shield tunnel crack as the research object, and proposed a dynamic block method to calculate the length and maximum width of the crack disease, and used the difference box dimension method to calculate the fractal dimension of cracks, which can better describe the characteristics of crack distribution and strike.

It can be seen from the above research that from the initial extraction of the two most basic parameters of crack width and length to the introduction of fractal dimension, block size and other parameters at this stage, it can be seen that the extraction of tunnel crack parameters has experienced considerable development [9], providing more accurate parameter support for the reliability evaluation of tunnel service. At present, the service reliability evaluation of tunnel mostly summarizes a set of corresponding evaluation system based on different parameters. The severity of tunnel diseases is classified in the evaluation system. However, the development status at home and abroad is different at this stage.

The development of tunnel safety grade evaluation in foreign countries is earlier, especially in the United States and Japan. In 2005, the United States issued the highway and railway traffic tunnel inspection manual [10], which divides the tunnel into four parts: tunnel structure, equipment system, power system and other systems. The rating of tunnel structure is divided into 10 levels, and the rating of other structures is divided into 5 levels. After extracting the parameters, Shigeta et al. [3] proposed a set of Tunnel Crack Index (TCI). This method is widely used in Japan. On the basis of it, Japan has edited relevant evaluation systems such as railway civil structures maintenance and management standards same interpretation (tunnel Edition) [11] and highway tunnel maintenance and management overview [12]. Different soundness standards are adopted at different stages, which divides the soundness into class A, class B Grade C and grade s, of which grade A is divided into Grade AA, grade A1 and grade A2.

Compared with the bridge and road maintenance specifications, China's highway tunnel maintenance specifications started relatively late. Before 2003, China had no relevant specifications applicable to tunnel maintenance, but some contents related to tunnel maintenance were given in the technical specifications for highway maintenance (JTG 073–96) issued in 1996. Then, a small part of China drew lessons from Japanese specifications and put forward the first highway tunnel specification, technical specification for highway tunnel maintenance (JTG H12-2003), which defined the inspection method and evaluation grade classification of highway tunnel civil structure. Then in 2015, China put forward a new maintenance specification,

technical specification for highway tunnel maintenance (JTG H 12–2015) [13], emphasizing the importance of other facilities to tunnel safety.

Domestic scholars have also done a lot of research on the safety classification of highway tunnels. Feng Xiaoyan [14] based on the investigation results of a large number of tunnel disease projects at home and abroad, studied and discussed the structural diseases of railway and highway tunnels in Japan and China, and proposed a comprehensive classification method for tunnel lining diseases based on the code for design of concrete structures, using the width of tunnel cracks as the evaluation index, so as to divide the cracks into four grades A, B, C and D. Zhang Yujun [15], Li Zhiguo et al. [16] graded the influence of crack depth, width, number of cracks and other factors on the safety of tunnel structure for the tunnel structure with reinforced concrete secondary lining. Luo Xin [17] based on the basic theory of analytic hierarchy process, constructed a highway tunnel health evaluation and judgment system. Combined with the special inspection data of domestic highway tunnel structures, the highway tunnel diseases are divided into four safety levels. Combined with the characteristics of the evaluation indexes, the weights of each index are established by using the product scale, fuzzy mathematics and artificial neural network, and then combined with the fuzzy mathematics evaluation method, the comprehensive evaluation model of the safety status of highway tunnel is determined, and the corresponding safety status evaluation system is developed. Zheng Lihuang [18] took the geometric properties of cracks such as crack density, spacing and stagger distance as the safety evaluation index to quantify and grade the lining cracks, and classified the lining cracks according to the horizontal and vertical stagger distance of the cracks.

Wu Senyang et al. [19] added risk parameters such as pavement structure and vehicle speed on the basis of European road tunnel operation risk parameters, which not only expanded the evaluation scope, but also improved the evaluation accuracy. Li [20], Zhu Hehua et al. [21] proposed the Tunnel Serviceability Index (TSI) to divide the tunnel into sections and use the evaluation index to evaluate the current operation safety status of each section of the tunnel. This method has good objectivity and uses different colors to express the current tunnel service status, which has strong visibility. Li Qingtong et al. [8] proposed the shield tunnel crack diagnosis index TDI-C on the basis of TSI, and established the classification standard of disease grade. This method was applied to a shield metro tunnel section in Shanghai. The practice shows that the proposed method can better describe the disease grade corresponding to the shield segment, and has the advantages of strong visibility and objective evaluation method.

The above scholars have done a lot of research on the tunnel safety rating method, but most of the research at this stage is aimed at the shield metro tunnel. This is because the shield tunnel is composed of a ring of segments, and its quantitative parameters and disease statistics are relatively simple. The evaluation on the lining service reliability of highway tunnel is relatively few and unsystematic, the classification suggestions given by the national standard method are relatively vague and abstract, which makes it difficult for maintenance personnel to actually classify. Therefore, for highway tunnel, it is very important to put forward a method that can quantitatively analyze the safety state of lining at this stage.

## 2. Crack parameter extraction

The crack parameter extraction method usually determines the physical parameters such as the length, width, aspect ratio and area of the circumscribed rectangle according to the number of pixels occupied by the rectangle when finding the smallest circumscribed rectangle of the crack. However, some scholars pointed out that it is inaccurate and unreasonable to determine the fracture severity only by physical parameters. Therefore, based on previous studies, this paper extracts three parameters: fracture length, fracture width and fractal dimension.

## 2.1 Obtaining tunnel crack binary image

The research object of this paper is the binary image of highway tunnel cracks. The method of obtaining the binary image of highway tunnel cracks is proposed to use the crack recognition method proposed in document [22]. The steps of obtaining the binary image are as follows: first, Relying on the deep learning framework CAFFE (Convolutional architecture for fast feature embedding) [23] developed by the vision and learning center of the University of California, Berkeley, a full convolution network structure suitable for lining crack disease image recognition is constructed, as shown in Fig 1; Secondly, the original image is segmented by the deep learning model of full convolution neural network; Finally, the binary image of crack disease is obtained. The binary image of highway tunnel crack obtained is shown in Fig 2.

## 2.2 Crack skeleton

Before extracting the fracture parameters, it is necessary to skeleton the fracture. Due to the particularity of the crack image, there may be a single pixel discontinuity in the identified crack image. Therefore, this paper uses an iterative algorithm to skeleton the crack image [24].

The crack skeleton after refinement is shown in Fig 3. It can be seen from Fig 3 that the extracted crack skeleton conforms to the basic shape of the crack, the overall trend and the inflection point of the crack have not changed, and there are almost no burrs and other phenomena.

## 2.3 Determination of crack length parameters

To extract the length of the skeletonized crack, first find the minimum circumscribed rectangle of the skeletonized crack. If the crack image contains two or more cracks, extract the minimum circumscribed rectangle respectively. If a circumscribed rectangle contains other cracks, white pixels of other cracks are defined as black background. When another crack is extracted, its pixels remain the original white. The principle of crack length extraction is shown in Fig 4.

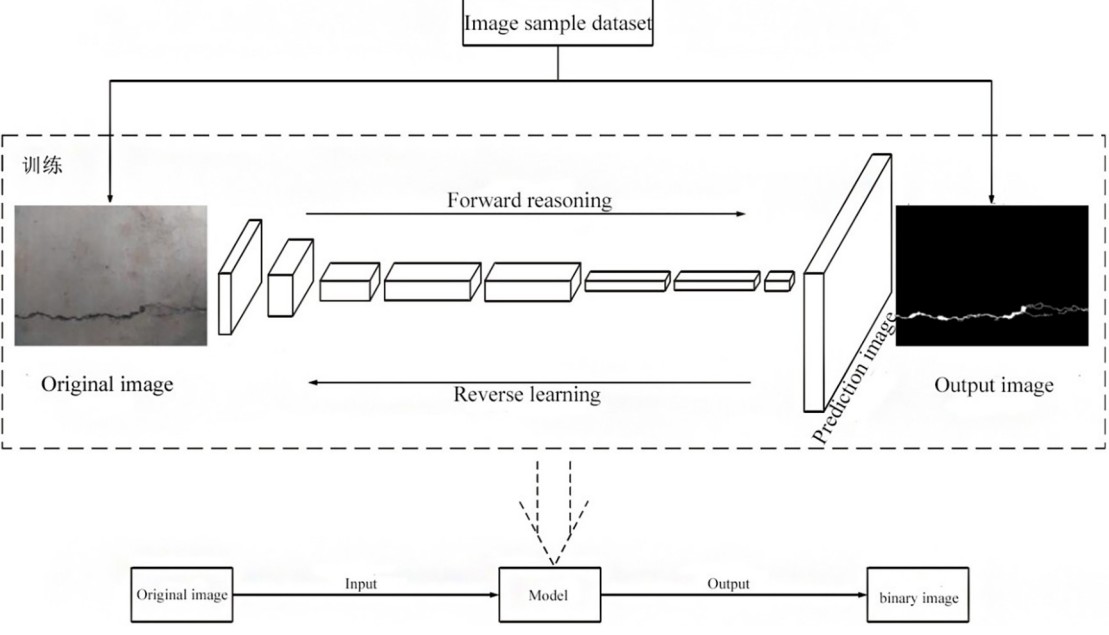

**Fig 1. Full convolutional network architecture for tunnel crack recognition.**

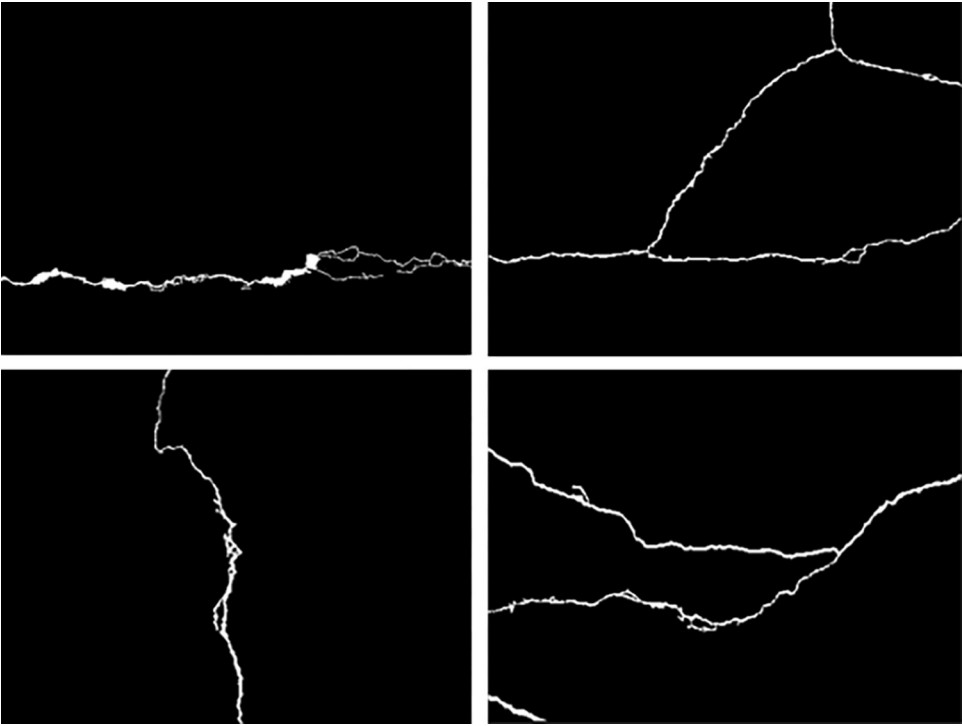

**Fig 2. Binary image of tunnel cracks.**

The specific steps are as follows:

(1) select the long side of the circumscribed rectangle as the length of the rectangle, divide the extracted minimum circumscribed rectangle by $n$, and the value of $n$ is divided by the pixel value of the long side of the circumscribed rectangle and the pixel value of the sub block.

(2) A rectangular coordinate system is established at the lower left corner of each divided fine block, and the distance $L_i$ between two crack points in the region is calculated according to Pythagorean theorem. $(x_i, y_i)$ is the intersection point between the crack and the left line of the fine block, $(x_{i+1}, y_{i+1})$ is the intersection point between the crack and the right line of block.

$$L_i = \sqrt{(x_{i+1} - x_i)^2 + (y_{i+1} - y_i)^2} \qquad (1)$$

(3) Add $L_i$ in each divided block. The calculated $L$ is the pixel length.

$$L = L_1 + L_2 + L_3 + \cdots + L_n \qquad (2)$$

(4) Multiply the calculated $L$ in (3) by the actual distance (0.5mm) corresponding to each pixel to obtain the true crack length $L_t$

$$L_t = 0.5 \times L \qquad (3)$$

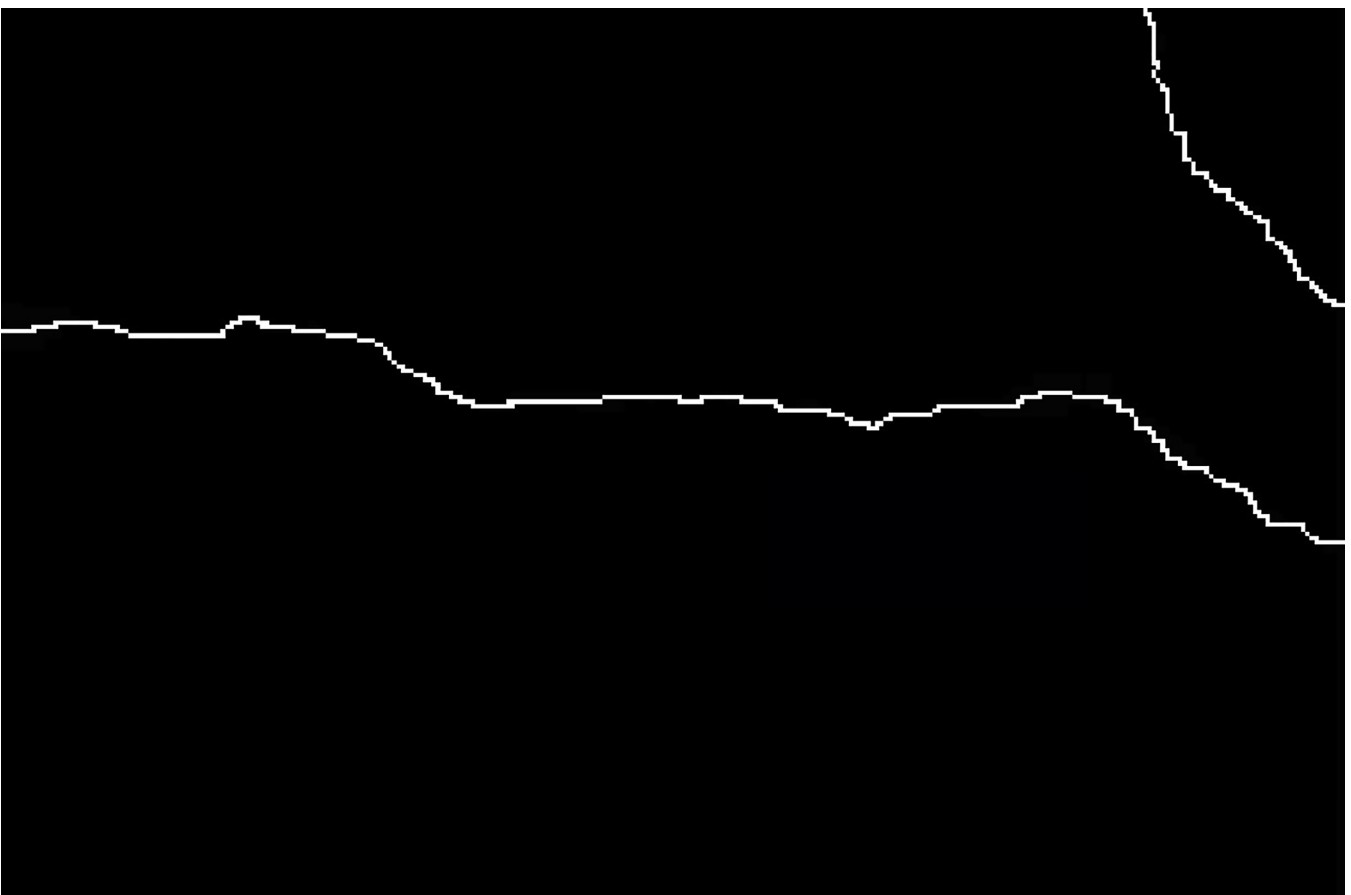

**Fig 3. Crack skeleton diagram.**

## 2.4 Extraction of maximum crack width parameters

The width of a crack may be different at different locations, and the depth of the crack at the position with the maximum width is usually the deepest, which is the position most in need of repair. Scholars pointed out in the research that the crack depth can be inferred from the crack width through the double line model [25], and the crack width is more convenient than the crack depth in the parameter extraction process. Therefore, the maximum crack width is selected as one of the parameters to be extracted in this paper.

The specific steps for extracting the maximum width are as follows:

1. Perform sliding window detection on the binarized image, and the detected image is shown in Fig 5.

2. The small rectangle that has been equally divided in the crack length extraction step corresponds to the sub block in the crack contour extraction in Fig 5;

3. In each small rectangle, the perimeter of the pixel points corresponding to the sub block in Fig 5 is marked as $P$, the total area of the pixel is marked as $Area$, and $P$ is the total length of the crack area calculated based on the eight neighborhood;

4. Convert the units of $Area$ and $P$ from pixels to millimeters;

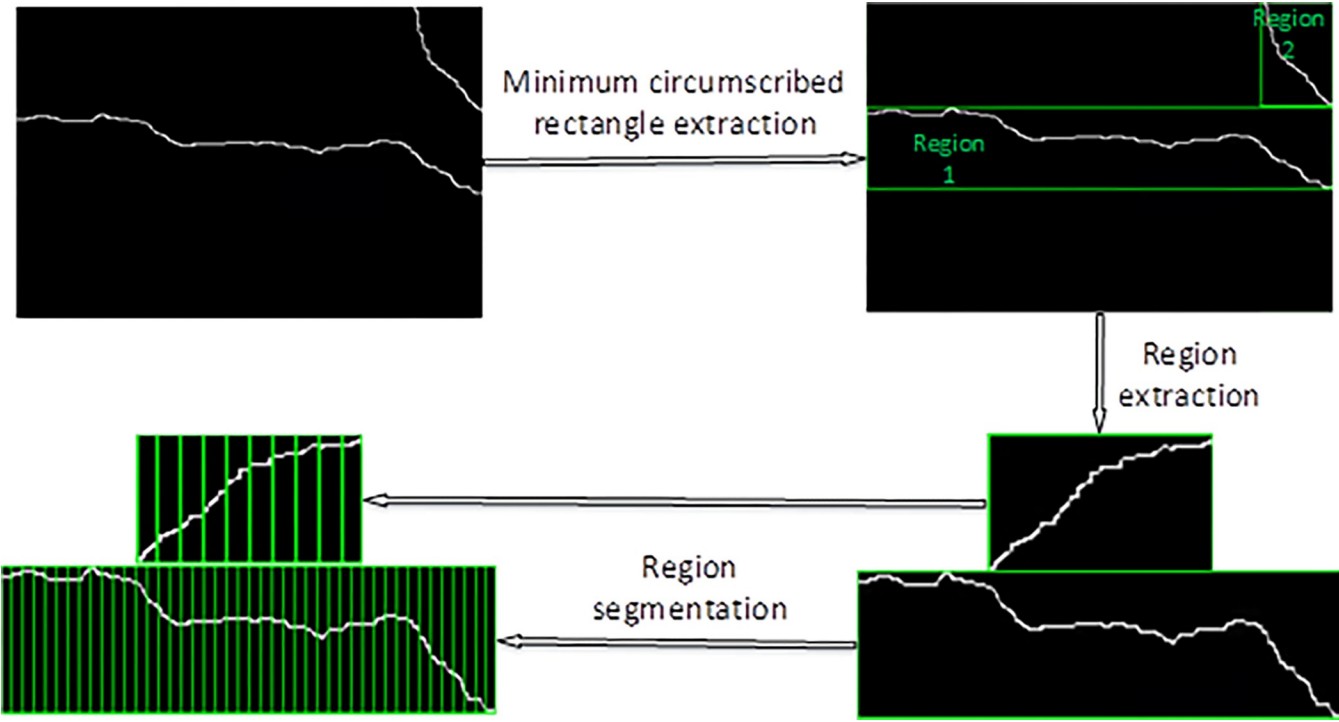

**Fig 4. Principle of crack length extraction.**

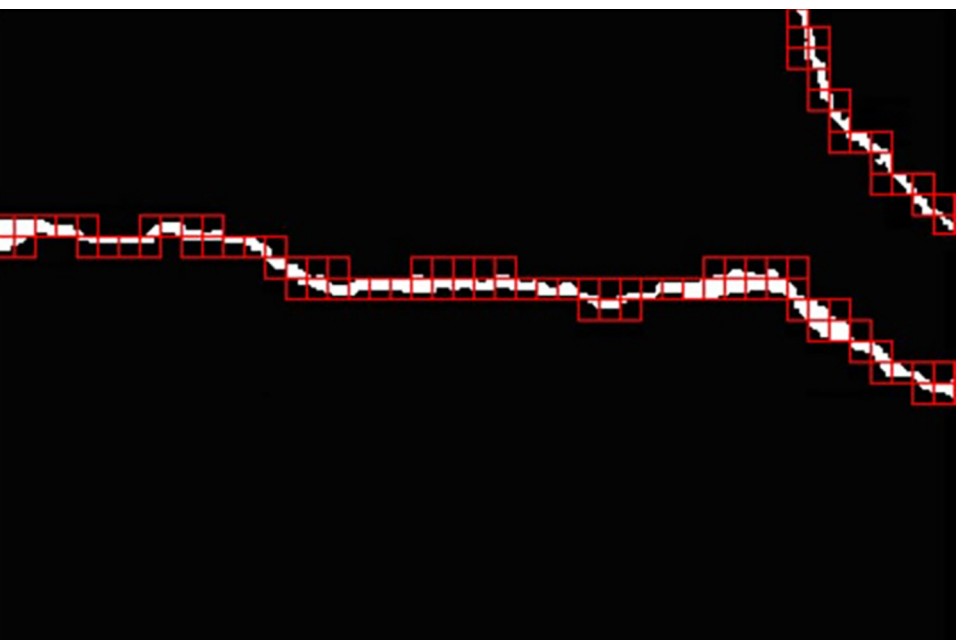

**Fig 5. Subimage division of cracks.**

5. Compare the true crack length $L_t$ with half of the crack perimeter $P$, and select the larger of the two as the true value;

6. The crack width $w_i$ is obtained by dividing the *Area* by the true value of the crack length;

7. The maximum crack width $W$ is the maximum of all $w_i$, i.e. $W = maxw_i$. The above steps are for extracting the maximum crack width.

## 2.5 Determination of fractal dimension

Fractal dimension belongs to the concept category of non Euclidean geometry. The main methods for calculating fractal dimension usually include box dimension method, capacity dimension method, similarity dimension method, random walk method, etc. [26, 27]. Compared with other fractal dimension calculation methods, box dimension has the characteristics of simple mathematical calculation and easy programming, so it is widely used in engineering practice. For the box fractal dimension calculation of a two-dimensional black-and-white image, after analyzing the calculation principle, Peng Ruidong [28] verified the effectiveness of the box fractal dimension mathematical calculation through four types of mathematical fractal structures with strict self similarity, such as *Cantor* triad, *koch* curve, rectangular *Sierpinski* gasket and triangular *Sierpinski* gasket. Therefore, the box dimension is used to calculate the fractal dimension of the binary image of tunnel lining surface cracks. FRACLAB, an external open source toolbox of Matlab, is used to calculate the fractal dimension of the image. Taking two common geometric figures of line and *koch* curve as examples, the fractal dimension of the binary image is calculated respectively, and the error is estimated. For the straight line, the theoretical value of box dimension is 1.0, the calculated value of toolbox is 1.0053, and the error is 0.53%; For *koch* curve, the theoretical value of box dimension is 1.2618, and the calculated value using the toolbox is 1.2708, with an error of 0.72%. The error calculation shows that the error between the actual value and the theoretical value of the fractal dimension calculated by the external toolbox in this paper is relatively small, no more than 1%, so the toolbox can be used to calculate the fractal dimension of tunnel cracks.

Because the calculation of fractal dimension requires only one crack in each crack image, the image needs to be processed first. First, keep region 1, change all white pixels in region 2 into black pixels, and get the crack diagram as shown in Fig 6(A). Then use FRACLAB to calculate its fractal dimension, as shown in Fig 6(B), and the box dimension is 1.1084.

By calculating the fractal dimensions of the four images in Fig 2, it can be seen that the cracks in Fig 7(A) are relatively simple, and their fractal dimensions are relatively small, which is 1.1757; In Fig 7(B), the crack is vertical, but there are more crack corners than (a), so the fractal dimension is relatively large, which is 1.1922; Fig 7(C) the fracture morphology is relatively complex, and the fractal dimension is 1.2496; The fracture in Fig 7(D) is roughly Y-shaped, with a fractal dimension of 1.3057. It can be found from the comparison in Fig 7 that the more complex the fracture shape is, the larger the fractal dimension is. Therefore, we can think that the box dimension can better judge the severity of fractures with different shapes and trends.

## 3. Establishment of disease sample space

### 3.1 Selection of diagnostic scale

Diagnostic scale refers to the selected diagnostic range for tunnel crack diagnosis, which can be an image, a crack or a tunnel section. The selected diagnostic scale must be applicable and conform to the real situation, and the situation that the selected diagnostic scale cannot describe the reliability of the tunnel section cannot occur. Section tunnel can avoid the situation that one image cannot contain the whole crack, and it can also avoid the situation that one crack cannot accurately reflect the reliability of tunnel lining structure. Therefore, the diagnostic scale selected in this paper is section tunnel.

*The technical code for maintenance of highway tunnels* (JTG H 12–2015) issued by the Ministry of transport of the people's Republic of China classifies highway tunnels according to

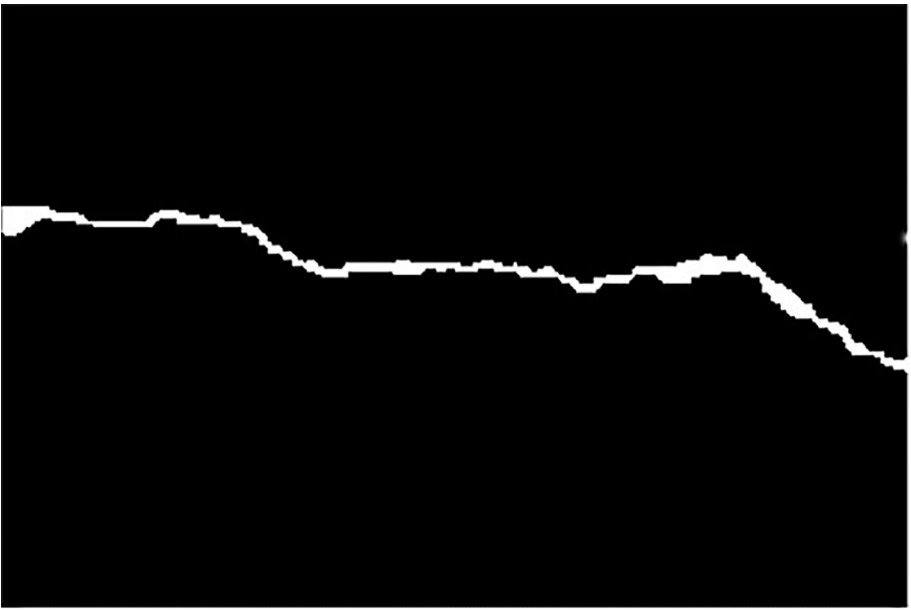

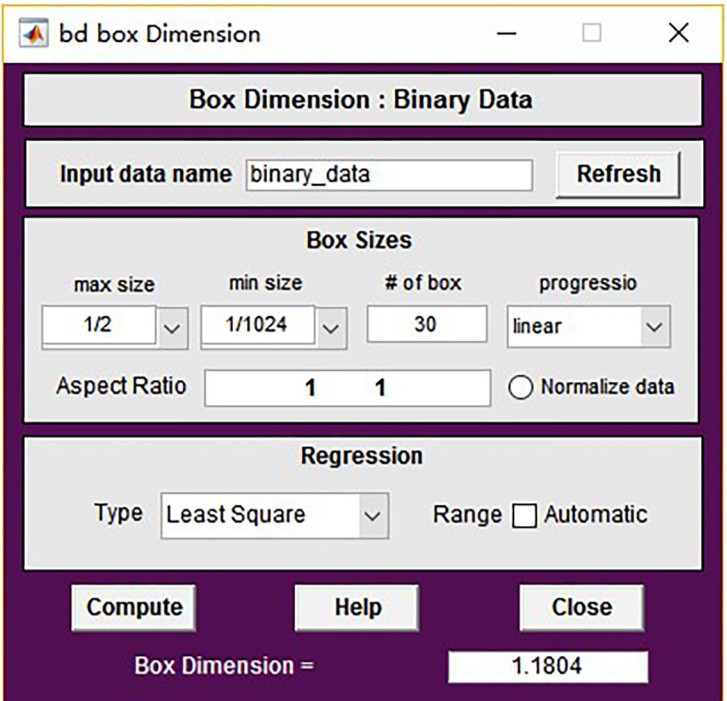

**Fig 6. Result of calculation of the fractal dimension of cracks.** (a) Crack binary diagram, (b) Fractal dimension calculation.

their length, in which the total length $L \leq 500m$ of the tunnel is short, $500m < L \leq 1000m$ is medium tunnel, $1000m < L \leq 3000m$ is long tunnel, and $3000m < L$ is super long tunnel.

In practical engineering, the tunnel length is usually not an integer, but it is also necessary to divide the tunnels of different lengths according to a certain length index. How to select the appropriate length index to divide the tunnels according to the length is the key to the division of tunnel sections. In document [15], 200 ring shield segments are taken as a section, but the shield

**Fig 7. Calculation of fractal dimension of fracture with different cracks.** (a)*FW* = 1.1757, (b) *FW* = 1.1922, (c) *FW* = 1.2496, (d) *FW* = 1.3057.

metro tunnel is selected, and the number of segments can be better quantified. However, highway tunnels and shield tunnels are different. If tunnels of different lengths are divided equally according to a certain number, the lengths of different tunnel sections are different, which will inevitably affect the safety grade of the evaluated section tunnel. Therefore, based on the *code for technical maintenance of highway tunnels*, this paper plans to divide 500*m* into a tunnel section.

Each highway tunnel has a central stake number. 500*m* is the first tunnel section divided by moving from a small stake number to a large stake number. If the total length of the tunnel is less than 500*m*, the whole tunnel is regarded as a section. Fig 8 is the schematic diagram of tunnel block division. In the figure, *A* is the standard tunnel section, with a length of 500*m*; *B* is a non-standard section tunnel with a length less than 500*m*.

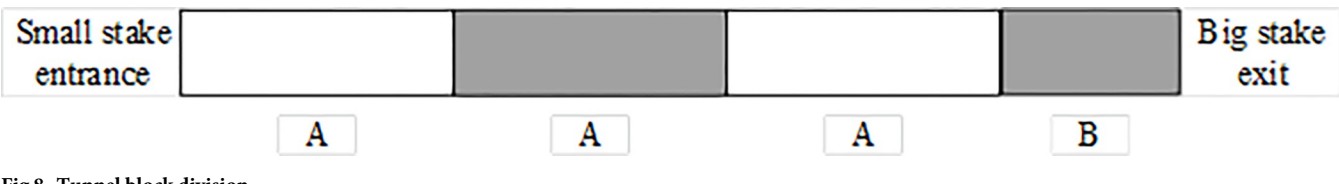

**Fig 8. Tunnel block division.**

## 3.2 Disease sample space

Before establishing the sample space, the digital characteristics of the quantized parameters should be determined first. The digital characteristics of the quantized parameters mainly include the average value, maximum value and cumulative value.

If the average value is selected as the digital feature of the quantitative parameter, the overall data will be averaged. What is displayed is the average level of the overall situation of the fracture, which will make the parameters extracted when the fracture is more complex smaller as a whole, and also make the parameters of the small fracture larger as a whole. If the maximum value is selected as the numerical characteristic of the quantitative parameter, it is the most unfavorable crack, and the influence of the number of cracks on the lining structure is not taken into account. For the cumulative value, it can well avoid the defects of the above two digital features, and can accumulate all the crack severity of the section tunnel, taking into account not only the maximum value of the crack parameters, but also the number of cracks. Therefore, this paper selects the cumulative value as the digital feature of the quantization parameter.

After determining the cumulative value as the evaluation feature of tunnel lining safety, it is necessary to establish the tunnel crack disease sample space. Based on the annual tunnel inspection in Anhui Province, this paper extracts the crack length, maximum width and fractal dimension parameters, and establishes the disease sample space. The sample space contains 200 tunnels, including 122 short tunnels, 66 medium tunnels, 10 long tunnels and 2 super long tunnels. A total of 280 sample spaces are obtained by extracting the parameters of each fracture section. The tunnel section with section length $B$ shall be corrected according to Formula (4).

$$I' = \frac{I}{B} \times A \qquad (4)$$

Where: $I'$ is the corrected tunnel crack parameter. $I$ extract the original values of the last three parameters for the fracture, including the fracture length, maximum width and fractal dimension.

For the obtained tunnel crack sample space, select 23 typical sample spaces and add a space with all 0 for comparison. The disease sample space is shown in Table 1.

**Table 1. Disease sample space of section tunnels.**

| Sample number | Cumulative crack length /mm | Cumulative crack width /mm | Fractal dimension | Sample number | Cumulative crack length /mm | Cumulative crack width /mm | Fractal dimension |
|---|---|---|---|---|---|---|---|
| 1 | 0 | 0 | 0 | 13 | 9562.27 | 31.04 | 32.26 |
| 2 | 4856.21 | 87.24 | 33.87 | 14 | 23268.51 | 59.62 | 61.05 |
| 3 | 29563.12 | 342.39 | 74.65 | 15 | 9059.37 | 30.27 | 28.53 |
| 4 | 10259.47 | 202.27 | 36.28 | 16 | 68421.45 | 146.66 | 201.28 |
| 5 | 50012.69 | 762.31 | 106.24 | 17 | 12657.21 | 42.59 | 46.59 |
| 6 | 8562.95 | 25.16 | 25.62 | 18 | 82165.34 | 220.67 | 280.41 |
| 7 | 15624.06 | 28.63 | 35.29 | 19 | 50160.20 | 152.31 | 150.32 |
| 8 | 59846.67 | 150.58 | 95.26 | 20 | 19562.32 | 66.93 | 61.24 |
| 9 | 40059.64 | 95.67 | 72.64 | 21 | 12236.59 | 36.55 | 33.37 |
| 10 | 41592.58 | 122.26 | 88.26 | 22 | 5980.24 | 20.26 | 34.42 |
| 11 | 60598.25 | 152.27 | 130.29 | 23 | 23582.64 | 69.30 | 79.27 |
| 12 | 31865.27 | 66.34 | 52.27 | 24 | 6852.27 | 28.53 | 44.40 |

## 4. Service reliability grade classification of tunnel lining

The number of grades is the quantitative grade that describes the disease degree during lining evaluation. When determining the number of disease grading files, it should be noted that if there are many grading files, the difference between different disease degrees is small, and the reflected disease degree deviates from the actual value; If the number of grading grades is small, the number of serious disease grades will be classified into the number of disease grades that can not be repaired, resulting in further deterioration of the disease section and the occurrence of safety accidents.

At the present stage, many scholars and codes have divided the number of grades of tunnels into different categories. For example, in the industrial standard technical standard for structural maintenance of urban rail transit tunnels (CJJ/T 289–2018) [29] edited by the Ministry of housing and urban rural development and Tongji University, the health standard is divided into five levels, i.e. health level 1 to health level 5. Health level 1 is a problem free tunnel, and health level 5 requires immediate closed maintenance. Article 2 of article 5.3 of the technical code for maintenance and repair of urban rail transit facilities (DB11/T 718–2016) [30], a local standard of Beijing issued by the Beijing Municipal Bureau of quality and technical supervision, points out that the status assessment of the tunnel is divided into five levels: Level 1 is a slight disease, level 2 is a medium disease, level 3 is a serious disease, level 4 is a serious disease, and level 5 is a very serious disease. According to table 3.3.3 [31] classification standard for service status of structural sections in the *code for service performance appraisal of shield tunneling structures* issued by Shanghai building materials industry market management station, the service status is divided into five levels: normal, degraded, inferior, deteriorated and hazardous. At this stage, the number of disease classification files is mostly qualitative, lacking relevant theoretical basis. Therefore, this paper uses EM clustering algorithm, which is commonly used in machine learning, to cluster 24 samples in the disease sample space, so as to reasonably determine the number of disease classification files. The clustering algorithm uses Euclidean distance as the evaluation index, that is, the closer the Euclidean distance between two points in the sample points, the higher the similarity of the samples. Such data are considered to belong to a cluster.

Silhouette Coefficient is an evaluation method of clustering effect. It combines cohesion and resolution. The main function of contour coefficient is to evaluate the results of different clustering algorithms or the number of grades corresponding to the same clustering algorithm when the data sets are the same. In Matlab, the contour coefficient is calculated by using the *silhouette* function [32]. Generally, the value of the contour coefficient is [−1,1]. If the value is closer to -1, it means that the clustering effect is worse. On the contrary, if the value is closer to 1, it means that the clustering effect is better. The calculation principle of contour coefficient is shown in Formula (5).

$$S_i = \frac{b_i - a_i}{max(a_i, b_i)}, i = 1, 2, 3, \cdots, n \tag{5}$$

Where: $a_i$ is the average distance from the sample $i$ to other samples in the same cluster. $b_i$ is the average distance from the sample $i$ to all samples of other clusters. $n$ is the sum of samples, n = 24 in this paper.

Fig 9 shows the average contour value under different levels. It can be seen from the bar graph that when the number of levels is 5, the average contour value corresponding to the sample space is the largest, reaching 0.89. Therefore, this paper divides the crack disease level of highway tunnel into 5 levels. Table 2 reflects the contour values corresponding to 24 samples in the sample space when the number of grading files is 5.

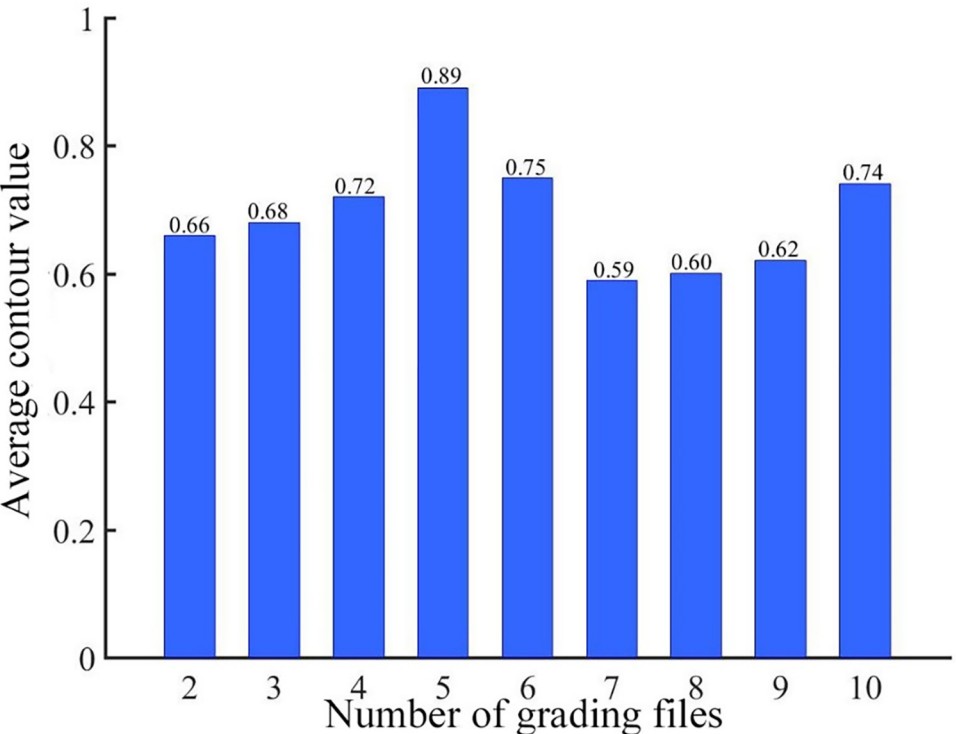

**Fig 9. The average silhouette values of different grades.**

It can be seen from Table 2 that when the number of grading grades is 5, the contour values of the sample space are greater than 0.7, which proves that the number of grading grades is 5 is reasonable. However, only the number of grades can not determine the disease degree of the sample space, so the relative European range radar chart shown in Fig 9 is drawn. The calculation formula of relative Euclidean distance D is shown in Formula (6). The radar chart shows the relative Euclidean distance between 23 sample points in the sample space and samples with all parameters of 0.

$$D = \sqrt{(M_i - M_0)(M_i - M_0)^T} \tag{6}$$

**Table 2. Sample space silhouette values when the number of grading is 5.**

| Sample number | Contour value | Sample number | Contour value |
|---|---|---|---|
| 1 | 0.89 | 13 | 0.95 |
| 2 | 0.90 | 14 | 0.88 |
| 3 | 0.90 | 15 | 0.87 |
| 4 | 0.72 | 16 | 0.89 |
| 5 | 0.91 | 17 | 0.89 |
| 6 | 0.88 | 18 | 0.90 |
| 7 | 0.89 | 19 | 0.91 |
| 8 | 0.90 | 20 | 0.87 |
| 9 | 0.92 | 21 | 0.88 |
| 10 | 0.88 | 22 | 0.90 |
| 11 | 0.90 | 23 | 0.93 |
| 12 | 0.93 | 24 | 0.92 |

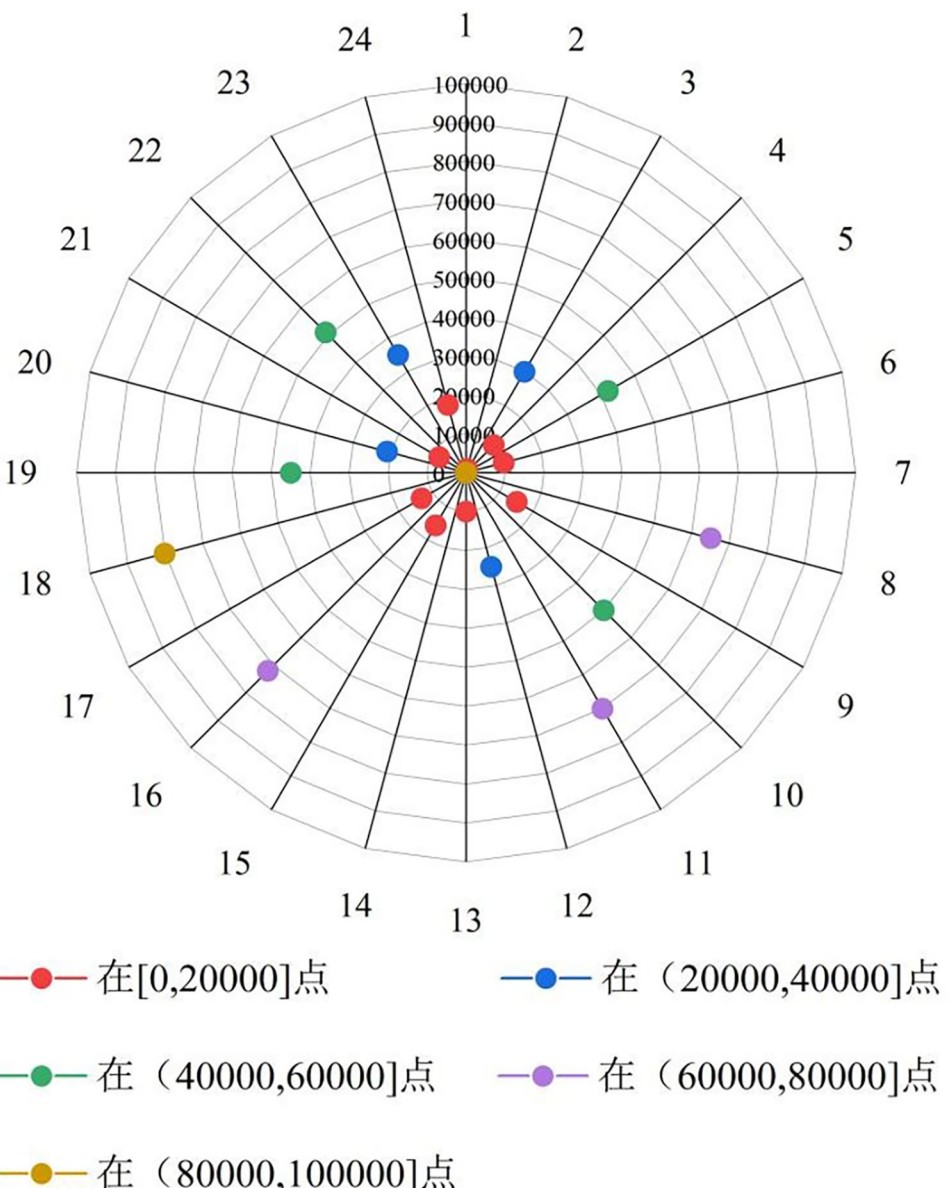

**Fig 10. Relative euclidean distance radar chart of sample points.**

Where: $M_i$ is a 1×3 matrix composed of three parameters of the ith sample in the disease sample space. In this paper, $i = 2,3,4,\cdots,24$。 $M_0$ is the sample in which all three parameters are 0.

It can be seen from Fig 10 that the relative Euclidean distances of sample points 2, 4, 6, 7, 9, 13, 15, 17, 21 and 24 are located in the interval [0, 20000], and sample points 3, 12 The relative Euclidean distances of No. 20 and No. 23 are in the interval (20000, 40000], the relative Euclidean distances of No. 5, No. 10, No. 19 and No. 22 of sample points are in the interval (40000, 60000], the relative Euclidean distances of No. 8, No. 11 and No. 16 of sample points are in the interval (60000, 80000], and the relative Euclidean distances of No. 18 of sample points are in the interval (80000, 100000]. In view of the fact that the number of disease grades is divided into five levels in this paper, combined with the relative Euclidean range radar map of the

**Table 3. Classification standard of highway tunnels.**

| Classification level | Disease description | Corresponding colors | Relative euclidean distance |
|---|---|---|---|
| Normal | The number of cracks is relatively small, indicating normal service | | [0, 20000] |
| Degraded | The number of cracks is small, which basically does not affect the normal service | | (20000, 40000] |
| Inferior | The number of cracks is large, which begins to affect normal service | | (40000, 60000] |
| Deteriorated | The number of cracks is large, which has affected the normal service | | (60000, 80000] |
| Hazardous | The number of cracks is large and serious, which affects the normal service of the structure | | (80000, 100000] |

sample points, the service status of the highway tunnel is divided into five levels: normal, degraded, inferior, deteriorated and hazardous.

The normal service state means that there are few cracks in the highway tunnel, which does not affect the normal service of the structure; The service state is degraded, which means that the number of cracks in the highway tunnel is small, but it basically does not affect the normal service state of the structure; The service state is inferior, which means that the number of cracks in the highway tunnel is large, and has begun to affect the normal service state of the structure; The service condition is deteriorated, which means that there are many cracks in the highway tunnel, which affects the normal service condition of the structure and requires timely maintenance; The service state is hazardous, which means that there are a lot of cracks in the highway tunnel, which affects the normal service of the structure and needs to immediately close the tunnel for maintenance. With reference to the traffic light management law, the highway tunnel classification standards are shown in Table 3.

## 5. Establishment of safety evaluation formula for tunnel lining

Euclidean distance method was used to determine the degree of disease represented by each grade. Based on this, it is very important to propose a quantitative method to analyze the reliability of tunnel lining, so partial least square method is adopted in this paper.

### 5.1 Mathematical principle of partial least square method

Partial Least Squares Regression (PLSR), also known as the partial least square method, is suitable for finding the basic relationship between two matrices, that is, an implicit variable method for modeling the covariance structure in these two spaces. The partial least squares model will try to find the multidimensional direction in $X$ space to explain the multidimensional direction with the largest variance in $Y$ space. Partial least squares regression is especially suitable when the prediction matrix has more variables than the observation, and there is multicollinearity in the value of $X$. Find a linear regression model by projecting the predicted and observed variables into a new space [33–35].

The basic idea to realize the partial least squares method is to maximize the covariance of $t_1$ and $u_1$, that is, to solve the optimization problem as shown in Formulas (7) and (8):

$$max\{cov(t_1, u_1)\} = max\langle E_0 w_1, F_0 c_1\rangle \tag{7}$$

$$s.t\begin{cases} w_1{}^T w_1 = 1 \\ c_1{}^T c_1 = 1 \end{cases} \tag{8}$$

Where: $E_0$ is the normalized data of $X$. $F_0$ is the standardized data of $Y$. $w_1$ is the unit eigenvector of $E_0{}^T F_0 F_0{}^T E_0$. $c_1$ is the unit eigenvector of $F_0{}^T E_0 E_0{}^T F_0$.

Calculate $w_1$ and $c_1$ to obtain components $t_1$ and $u_1$, and then calculate the regression formulas of $E_0$ and $F_0$ for $t_1$ respectively. The specific mathematical expression of the regression formula is shown in Formulas (9) and (10).

$$E_0 = t_1 p_1{}^T + E_1 \tag{9}$$

$$F_0 = t_1 r_1{}^T + F_1 \tag{10}$$

Where: $p_1$ is the regression coefficient vector, $p_1 = E_0{}^T t_1/\|t_1\|^2$; $r_1$ is the regression coefficient vector, $r_1 = F_0{}^T t_1/\|t_1\|^2$; $E_1$、 $F_1$ is the residual matrix of the regression formula.

Replace $E_1$、 $F_1$ with residual matrix $E_0$、 $F_0$. After cyclic calculation, the formula shown in Formula (11) is finally obtained.

$$F_0 = t_1 r_1{}^T + t_2 r_2{}^T + \cdots + t_A r_A{}^T + F_A = E_0\left[\sum\nolimits_{j=1}^{A} w_j{}^{\bullet} r_j{}^T\right] + F_A \tag{11}$$

Where: $w_j{}^{\bullet} = \prod_{i=1}^{j-1}(I - w_i p_i{}^T)w_j$, $\sum_{j=1}^{A} w_j{}^{\bullet} r_j{}^T$ are partial least squares coefficient vectors; A is the rank of X space.

## 5.2 Establishment of index calculation formula

Partial least square method has been applied in practical research by many scholars. Li et al. [20] proposed the tunnel applicability index TSI through partial least square regression, and gave the result distribution of TSI by taking Shanghai metro tunnel section as an example. Because it has been proved that it can be well applied to various index calculation formulas, this paper uses the partial least square method to calculate the Highway tunnel Service Reliability (HSR) calculation formula.

In this paper, three input parameters are set as crack length, maximum width and fractal dimension, and a 3×24 as the independent variable matrix of PLSR; The relative Euclidean distance of the sample is constructed into a size of 1×24 as the dependent variable matrix of PLSR, as shown in the third and eighth columns of Table 4. Then construct a partial least

**Table 4. Calculation of various indicators of *HSR*.**

| Sample number | Reliability level and corresponding color | Relative euclidean distance | *HSR* calculated | Residual | Sample number | Reliability level and corresponding color | Relative euclidean distance | *HSR* calculated | Residual |
|---|---|---|---|---|---|---|---|---|---|
| 1 | Normal | 0.00 | 375.47 | 375.47 | 13 | Normal | 9562.37 | 9584.27 | 21.90 |
| 2 | Normal | 4857.11 | 4925.90 | 68.79 | 14 | Degraded | 23268.67 | 23186.05 | -82.62 |
| 3 | Degraded | 29565.20 | 29543.32 | -21.88 | 15 | Normal | 9059.47 | 9114.85 | 55.38 |
| 4 | Normal | 10261.33 | 10433.34 | 172.01 | 16 | Deteriorated | 68421.90 | 68479.58 | 57.68 |
| 5 | Inferior | 50018.61 | 50100.98 | 82.37 | 17 | Normal | 12657.37 | 12608.35 | -49.02 |
| 6 | Normal | 8563.04 | 8404.90 | -158.14 | 18 | Hazardous | 82265.83 | 82414.81 | 148.98 |
| 7 | Normal | 15624.14 | 15730.03 | 105.89 | 19 | Inferior | 50160.66 | 50061.54 | -99.12 |
| 8 | Deteriorated | 59846.95 | 59707.35 | -139.60 | 20 | Degraded | 19562.53 | 19514.37 | -48.16 |
| 9 | Inferior | 40059.81 | 39978.28 | -81.53 | 21 | Normal | 12236.69 | 12253.14 | 16.45 |
| 10 | Inferior | 41592.89 | 41553.41 | -39.48 | 22 | Normal | 5980.37 | 6003.70 | 23.33 |
| 11 | Deteriorated | 60598.57 | 60538.67 | -59.90 | 23 | Degraded | 23582.87 | 23552.32 | -30.55 |
| 12 | Degraded | 31865.38 | 31937.68 | 72.30 | 24 | Normal | 6852.45 | 6767.62 | -84.83 |

squares regression formula, as shown in Formula (12).

$$HSR' = \alpha L + \beta W + \lambda FW + C \tag{12}$$

In Formula (12), $L$ is the cumulative value of crack length; $W$ is the cumulative value of crack width; $FW$ is the cumulative value of fracture fractal dimension; $C$ is a constant. Each group of data in the disease sample space corresponds to the relative Euclidean distance. Use the *plsregress* command in *Matlab* to obtain the values of $\alpha$, $\beta$, $\lambda$ and $C$ in Formula (12). After calculation, in this paper, $\alpha = 0.9910$, $\beta = -0.7185$, $\lambda = 12.2290$, $C = -375.4654$. Therefore, Formula (12) in this paper can be transformed into Formula (13). The estimated value of $HSR$ is shown in the fourth and ninth columns of Table 4.

$$HSR' = 0.991L - 0.719W + 12.229FW - 375.465 \tag{13}$$

It should be noted that when the parameters $L$, $W$ and $FW$ are all 0, the obtained $HSR$ value is 0. Now transfer the constant $C$ on the right side of the formula to the left side of the formula to form the expression shown in Formula (14).

$$HSR = HSR' - C \tag{14}$$

The residual value is the difference between the fitted value and the true value, and it is also an important means to evaluate the *PLSR* regression model. The smaller the residual value, the better the fitting effect; On the contrary, the larger the residual value, the worse the fitting effect. Draw the residual diagram as shown in Fig 11. The residual values are shown in the fifth and tenth columns of Table 4.

It can be seen from Fig 11 that the error is less than 0.5% compared with the original relative Euclidean distance of 80100. The residual values of data points 2 to 24 are all in [-200200], and the average residual value is 5.26, indicating that the fitting effect of partial least square method is good. Due to the influence of initial parameter $C$, No. 1 data point has the largest residual value of 375.465, but it does not affect the validity of the formula. To sum up, the health

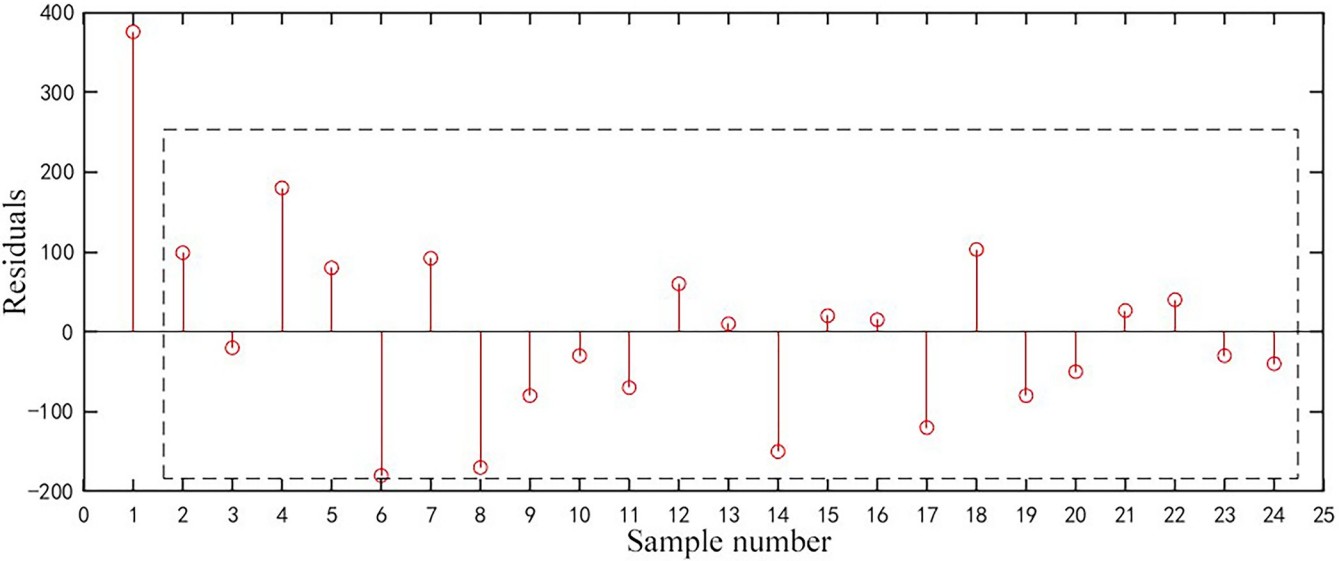

**Fig 11. Residuals corresponding to sample points.**

diagnosis formula of highway tunnel proposed in this paper has good applicability and can basically meet the needs of health diagnosis of highway tunnel at the present stage in China.

# 6. Comparison between engineering application and existing methods

## 6.1 Practical application

Tiantangzhai tunnel is located on G346 Shangan line, Jinzhai County, Liuan City, Anhui Province. Its central stake is *K745+228*. It is a single tunnel with a length of *2887m* and a net width of *10m*. It passed the completion acceptance on November 14, 2014. The tunnel mainly passes through the strongly and moderately weathered granite gneiss stratum, with developed joint fissures, relatively complete rock mass, hard lithology, and the maximum buried depth of about *218m*. The waterproof and drainage design of the tunnel is based on the principle of composite lining structure. The tunnel adopts composite pavement, which is consistent with the flexible pavement outside the tunnel. The inspection company conducted regular inspection on the tunnel on July 18, 2021, mainly focusing on the disease detection of the tunnel lining.

Identify the tunnel lining cracks obtained in the field inspection, and obtain the cumulative values of three parameters: crack length, maximum width and fractal dimension. The cumulative values of the three parameters are shown in the second, third and fourth columns of Table 8. As the total length of the tunnel is *2887m*, according to the crack section classification rules proposed in this paper, Tiantangzhai tunnel is divided into *6* sections, of which *5* sections are *500m* long and 1 section is *387m* long. The service reliability evaluation formula of highway tunnel proposed in this paper is used to evaluate the reliability of six section tunnels, and the service reliability diagram of section highway tunnel shown in Fig 12 is drawn according to the overall trend of the tunnel. It can be clearly seen from the figure that three tunnel sections are in the normal state defined in this paper, two tunnel sections are in the degraded state defined in this paper, and one tunnel section is in the degraded state defined in this paper. Fig 12 can visually show the health status of the section tunnel at the present stage, and provide an intuitive basis for the highway management and maintenance department to timely respond to the inferior, deteriorated and hazardous tunnel section and take necessary engineering measures to protect the normal operation of highway traffic and the personal and property safety of citizens.

## 6.2 Expert scoring method to evaluate the safety level of tunnel section

The expert scoring method refers to the method of sending a questionnaire to relevant experts anonymously, asking for the opinions of relevant experts, then recovering the questionnaire, then making statistics, induction and Analysis on the scores returned by experts, and

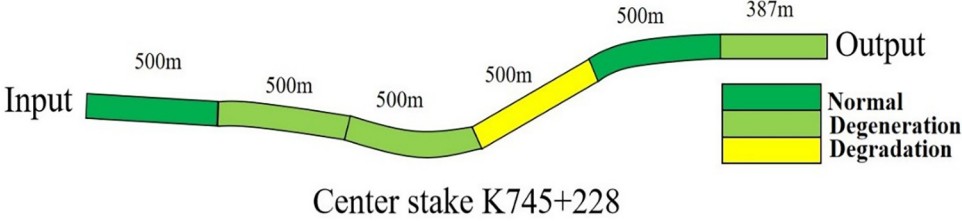

**Fig 12. Safety assessment of Tiantangzhai tunnel section.**

**Table 5. Specific scores of expert scoring method.**

| tunnel section / Questionnaire number | Section 1 | Section 2 | Section 3 | Section 4 | Section 5 | Section 6 |
|---|---|---|---|---|---|---|
| 1 | 1.3 | 2.1 | 2.5 | 2.9 | 1.2 | 2.3 |
| 2 | 1.5 | 1.9 | 2.7 | 3.5 | 1.3 | 2.4 |
| 3 | 1.2 | 2.0 | 2.4 | 3.2 | 1.5 | 2.2 |
| 4 | 1.3 | 1.8 | 2.1 | 3.6 | 0.9 | 2.0 |
| 5 | 1.0 | 2.0 | 2.0 | 3.0 | 1.0 | 2.0 |
| 6 | 0.8 | 2.5 | 2.0 | 3.5 | 1.3 | 1.8 |
| 7 | 1.2 | 2.7 | 2.1 | 3.8 | 1.5 | 2.4 |
| 8 | 1.3 | 2.2 | 2.2 | 3.7 | 1.4 | 2.3 |
| 9 | 1.2 | 2.6 | 2.0 | 3.3 | 1.6 | 2.5 |
| The average score | 1.2 | 2.2 | 2.2 | 3.4 | 1.3 | 2.2 |

combining the views of most experts on a certain problem to reasonably estimate some technical problems that are difficult to estimate quantitatively.

With the help of the internal platform of the testing company, this paper sent a questionnaire survey to 12 experts in its research institute. The 12 experts came from the road tunnel construction, monitoring, maintenance and new technology research and development directions. Before starting the road tunnel section safety evaluation, they informed the experts of the research method of this paper. During the scoring process, they were prohibited from communicating with each other. In view of the fact that the number of disease grading grades determined in this paper is 5, the expert rating is also divided into 5 grades, which are 1 point, 2 points, 3 points, 4 points and 5 points respectively. The corresponding safety levels are: normal, degraded, inferior, deteriorated and hazardous. The lower the score, the better the safety condition of the tunnel section. On the contrary, the higher the score, the worse the safety condition of the tunnel section. The questionnaires were collected and 9 valid questionnaires were obtained. The specific scores given by experts are shown in Table 5.

According to the average scores given by experts in Table 5, the average score of section 1 is 1.2, Section 2 is 2.2, section 3 is 2.2, section 4 is 3.4, section 5 is 1.3, and section 6 is 2.2, which basically corresponds to the tunnel safety evaluation method proposed in this paper.

### 6.3 Safety grade evaluation of tunnel section by national standard method

The national code method to evaluate the safety grade of tunnel section means to evaluate the safety of tunnel according to the existing national codes and regulations. In this paper, the Tiantangzhai tunnel is evaluated and analyzed according to the commonly used industry standard "technical specification for highway tunnel maintenance" (JTG H12-2015, hereinafter referred to as the technical specification).

The technical specification is applicable to the evaluation of highway tunnel by drilling and blasting method. The evaluation basis is the overall technical condition evaluation of civil structure in table 3.2.2 of the technical specification. The evaluation table is shown in Table 6.

The technical score of civil structure is shown in Formula (15).

$$JGCI = 100 \times \left[ 1 - \frac{1}{4} \sum_{i=1}^{n} \left( JCCI_i \times \frac{w_i}{\sum_{i=1}^{n} w_i} \right) \right] \tag{15}$$

Where: $w_i$ is the sub item weight; $JGCI_i$ Is the subitem status value, with a value range of 0~4.

**Table 6. Evaluation categories of overall technical status of highway tunnels [23].**

| Technical status judgment category | Description of civil structure assessment category |
|---|---|
| Class 1 | In perfect condition. There is no abnormal situation, or the abnormal situation is minor and has no impact on traffic safety |
| Class 2 | Slightly damaged. There is slight damage, and it is stable at this stage, which will not affect traffic safety |
| Class 3 | Moderate damage. There is damage and development is slow, which may affect pedestrian and traffic safety |
| Class 4 | Seriously damaged. There are serious damages, rapid development, and it is easy to affect the safety of pedestrians and traffic |
| Class 5 | Dangerous condition. There is serious damage, which has endangered the safety of pedestrians and traffic |

The calculation formula is shown in Formula (16).

$$JGCI_i = max(JGCI_{ij}) \tag{16}$$

Where: $JGCI_{ij}$ is the condition value of each sub item inspection section. $j$ is to check the paragraph number. The value is taken according to the actual number of segments. The value in this paper is 6.

Since this paper only calculates the lining cracks, it is assumed that all subdivisions of the civil structure are free of diseases, that is, the $w_i$ of the lining cracks is 100, and $JGCI_{ij}$ is deducted according to the deduction value in the technical code for highway tunnel maintenance. Since the 6 crack sections divided in this paper have cracks with a width greater than or equal to 1mm, the deduction value is 40 points, that is to say, the $JGCI_i$ value is 60. Since the value of $w_i$ in this paper is 100, the corresponding value of $n$ is 1. The scoring limit value of technical code for highway tunnel maintenance is shown in Table 7.

According to the calculation, the $JGCI$ value of the six tunnel sections in this paper is 85 points, that is, the corresponding category 1 in the technical condition assessment classification of civil structures.

## 6.4 Comparison of existing methods

The proposed safety evaluation method for tunnel section is compared with the expert scoring method and the national standard method. The comparison results are shown in Table 8.

From the comparison in Table 8, it is obvious that the expert scoring method can better distinguish the safety levels corresponding to the tunnel crack sections. However, compared with the method proposed in this paper, the expert scoring method is time-consuming and the evaluation criteria are not unified. If the national standard method evaluates the safety level of the tunnel section, it will make the difference of the safety level of the section not obvious, and the whole is more general and fuzzy, which is not conducive to the fine distinction of the safety level of each section. Firstly, the method proposed in this paper has good visibility. The safety grade of tunnel section is divided according to different colors, and the classification effect is good, which has high engineering application value.

**Table 7. Classification thresholds for assessment of technical status of civil structures.**

| Technical condition score | Assessment and classification of technical status of civil structures | | | | |
|---|---|---|---|---|---|
| | Class 1 | Class 2 | Class 3 | Class 4 | Class 5 |
| JGCI | ≥85 | ≥70, <85 | ≥55, <70 | ≥40, <55 | <40 |

**Table 8. Comparison of existing methods.**

| Tunnel section | Crack length (mm) | maximum width (mm) | Fractal dimension | *HSR* calculated | *HSR* Security level | Expert scoring method | National normative law |
|---|---|---|---|---|---|---|---|
| 1 | 6205.42 | 122.64 | 77.28 | 7006.51 | Normal | Normal | Class 1 |
| 2 | 20112.58 | 210.29 | 144.45 | 21546.95 | Degraded | Degraded | Class 1 |
| 3 | 21128.05 | 198.97 | 162.06 | 22776.77 | Degraded | Degraded | Class 1 |
| 4 | 40120.67 | 410.46 | 222.28 | 42182.93 | Inferior | Inferior | Class 1 |
| 5 | 3227.28 | 62.20 | 31.05 | 3533.25 | Normal | Normal | Class 1 |
| 6 | 19900.11 | 210.45 | 155.82 | 21475.32 | Degraded | Degraded | Class 1 |

## 7. Conclusions

With the rapid development of highway construction in China, the number and total length of highway tunnels are increasing. More and more tunnels have exposed a series of problems such as lining cracking in the long-term operation. In this paper, the service reliability evaluation method of highway tunnel is proposed. Through the application in practical engineering, the following conclusions are drawn:

1. Three parameters are extracted to evaluate the service reliability of highway tunnel lining, which are crack length, maximum width and fractal dimension. The extracted parameter values can quantitatively describe the tunnel crack behavior and severity to a certain extent, and lay a foundation for subsequent lining service reliability evaluation.

2. The diagnosis scale is selected as the section tunnel, 500m is taken as the section tunnel length, and the three extracted parameters are used to construct the disease sample space. The 23 samples in the sample space and the control samples are classified into 5 levels by EM clustering algorithm, which are normal, degraded, inferior, deteriorated and hazardous. The partial least square method is used to determine the service reliability evaluation formula of highway tunnel lining, and it is proved that the formula has a good fitting effect by calculating the residual value, which can be applied to the safety rating of highway tunnel lining.

3. The formula proposed in this paper is applied to engineering practice, and compared with the existing expert scoring method and national standard method. The results show that the classification method proposed in this paper has the advantages of high rationality, easy engineering practice and high safety grade discrimination.

## Supporting information

**S1 Data.**
(XLSX)

## Author Contributions

**Conceptualization:** Chunquan Dai.

**Data curation:** Zhaochen Zhou, Kun Jiang.

**Formal analysis:** Huidi Zhang.

**Funding acquisition:** Haisheng Li, Haiyang Yu.

**Investigation:** Zhaochen Zhou, Huidi Zhang.

**Methodology:** Kun Jiang.

**Project administration:** Chunquan Dai.

**Resources:** Chunquan Dai, Kun Jiang, Haisheng Li, Haiyang Yu.

**Software:** Zhaochen Zhou, Kun Jiang.

**Supervision:** Chunquan Dai.

**Validation:** Zhaochen Zhou, Huidi Zhang, Haisheng Li, Haiyang Yu.

**Visualization:** Zhaochen Zhou.

**Writing – original draft:** Zhaochen Zhou.

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
