## [Decision Letter · Decision Letter 0]

27 Mar 2023

PONE-D-22-32743Service Reliability Evaluation of Highway Tunnel Based on Digital Image ProcessingPLOS ONE

Dear Dr. Chunquan,

Thank you for submitting your manuscript to PLOS ONE. After careful consideration, we feel that it has merit but does not fully meet PLOS ONE’s publication criteria as it currently stands. Therefore, we invite you to submit a revised version of the manuscript that addresses the points raised during the review process.

ACADEMIC EDITOR: Please insert comments here and delete this placeholder text when finished. Be sure to:I just received comments from one reviewer.Please carefully revise your manuscript by addressing these comments.Clearly highlight the novelty of your study. ============================== Please submit your revised manuscript by May 11 2023 11:59PM. If you will need more time than this to complete your revisions, please reply to this message or contact the journal office at plosone@plos.org. Please include the following items when submitting your revised manuscript:A rebuttal letter that responds to each point raised by the academic editor and reviewer(s). You should upload this letter as a separate file labeled 'Response to Reviewers'.A marked-up copy of your manuscript that highlights changes made to the original version. You should upload this as a separate file labeled 'Revised Manuscript with Track Changes'.An unmarked version of your revised paper without tracked changes. You should upload this as a separate file labeled 'Manuscript'.

We look forward to receiving your revised manuscript.

Kind regards,

Jianguo Wang, PhD

Academic Editor

PLOS ONE

Journal Requirements:

"The authors declare that they have no known competing financial interests or personal relationships that could have appeared to influence the work reported in this paper."

Reviewers' comments:

Reviewer's Responses to Questions

**Comments to the Author**

1. Is the manuscript technically sound, and do the data support the conclusions?

Reviewer #1: Partly

2. Has the statistical analysis been performed appropriately and rigorously? 

Reviewer #1: Yes

3. Have the authors made all data underlying the findings in their manuscript fully available?

Reviewer #1: Yes

4. Is the manuscript presented in an intelligible fashion and written in standard English?

Reviewer #1: Yes

5. Review Comments to the Author

Reviewer #1: Dear editor,

3 features of cracks are extracted, EM clustering is conducted to obtain the safety grades, and the silhouette coefficient is used for evaluating the clustering effect. Partial least square method has been applied for estimating the Euclidean distance. The proposed method is compare with other method, and it shows the advantages of high rationality of the method. The work is fair and the contribution is satisfactory in Section 1-3. But Section 4 make me confused. Below, I proposed some enhancements necessary for this paper.

1. There are two same grades in 5 grades of tunnels’ safety grades in abstract.

2. 1.3 section has appeared for twice in your article, please check.

3. Parameters in 1.3 should be defined, such as .

4. In Section III, level 3 is same as level 4.

5. Before conducting the EM clustering, you had better to check your features’ range. Especially the distance-based algorithms, the feather with high range may have bigger influence on your results. Considering to standardize or normalize each features in Section III.

6. The disease description in your table 3, normal level should be in normal service instead of not in normal service.

7. It makes me confused that you conduct the PLSR to evaluate the tunnels’ safety after using EM clustering to obtain appropriate the levels of disease and using Euclidean distance range to explain it. Is It simpler to use your evaluation method than the computation of Euclidean distance? In your section 3, you can obtain the actual safety level of the tunnel by calculating the Euclidean distance of your variances. The Section 4 seems like no contribution to the others. Why is your Section 4 necessary for this article?

8. Your variances are the fracture length, maximum width and fractal dimension respectively before the Section 4, and variances become to cumulative value in Section 4. Please be consistent if you make mistakes of expression.

9. HSR is proposed as , so the HSR may be . Substituting the sample number 1’s features ( ) into HSR, I get whereas the HSR calculated result is 375.47 in table 4. It also makes me confused. Please explain it to me. The same questions also appear in table 8.

6. PLOS authors have the option to publish the peer review history of their article (what does this mean?). If published, this will include your full peer review and any attached files.

Reviewer #1: No

---

## [Author Response · Author response to Decision Letter 0]

27 Apr 2023

Firstly, we would like to thank you for your kind letter and for reviewers’ constructive comments concerning our article (Manuscript No.:PONE-D-22-32743).These comments are all valuable and helpful for improving our article. All the authors have seriously discussed about all these comments. According to the reviewers’ comments, we have tried best to modify our manuscript to meet with the requirements of your journal. In this revised version, changes to our manuscript within the document were all highlighted by using red colored text. Point-by-point responses to the reviewers are listed below this letter.

1. There are two same grades in 5 grades of tunnels’ safety grades in abstract.

In view of this problem, the safety grades mentioned in the whole paper are revised and uniformly defined as normal, degraded, inferior, deteriorated and hazardous.

2. 1.3 section has appeared for twice in your article, please check.

In response to this issue, the section numbering in the article has been revised.

3. Parameters in 1.3 should be defined, such as .

The parameters x_i、y_i, which were not mentioned in section 1.3, have been defined. (x_i,y_i) is the intersection point between the crack and the left line of the fine block, (x_(i+1),y_(i+1)) is the intersection point between the crack and the right line of block.

4. In Section III, level 3 is same as level 4.

In view of this problem, the safety grades mentioned in the whole paper are revised and uniformly defined as normal, degraded, inferior, deteriorated and hazardous.

5. Before conducting the EM clustering, you had better to check your features’ range. Especially the distance-based algorithms, the feather with high range may have bigger influence on your results. Considering to standardize or normalize each features in Section III.

This problem has not been modified because the disease data of tunnel samples is real data with a large variation between samples. If standardized or normalized processing is performed, the safety classification will be inaccurate due to the reduced variation between samples.

6. The disease description in your table 3, normal level should be in normal service instead of not in normal service.

There is a writing error here. The correct sentence should be "The number of cracks is relatively small, indicating normal service."

7. It makes me confused that you conduct the PLSR to evaluate the tunnels’ safety after using EM clustering to obtain appropriate the levels of disease and using Euclidean distance range to explain it. Is It simpler to use your evaluation method than the computation of Euclidean distance? In your section 3, you can obtain the actual safety level of the tunnel by calculating the Euclidean distance of your variances. The Section 4 seems like no contribution to the others. Why is your Section 4 necessary for this article?

The third section categorizes the safety level of tunnels using clustering methods, but it does not provide a description of the degree of damage of the samples. Instead, the degree of damage is differentiated by calculating the Euclidean distance between samples. In the fourth section, a reliability evaluation formula is proposed for tunnel linings, based on the determined classification levels and the corresponding degree of damage.

The issue has been corrected in the original text, with an additional paragraph added to the fourth section to provide an explanation.

8. Your variances are the fracture length, maximum width and fractal dimension respectively before the Section 4, and variances become to cumulative value in Section 4. Please be consistent if you make mistakes of expression.

This article uses cumulative values, and the crack length, maximum width, and fractal dimension in Table 1 are all cumulative values of tunnel defect samples. To avoid any misunderstanding, the original Table 1 has been modified, with the crack length and maximum width changed to "cumulative crack length" and "cumulative crack width," respectively.

9. HSR is proposed as , so the HSR may be . Substituting the sample number 1’s features ( ) into HSR, I get whereas the HSR calculated result is 375.47 in table 4. It also makes me confused. Please explain it to me. The same questions also appear in table 8.

The partial least squares regression equation constructed in this article is HSR'=αL+βW+λFW+C. The constant term C has a numerical value of 375.456, which has been rounded to 375.47 in the table to maintain consistency in the format.

---

## [Decision Letter · Decision Letter 1]

2 Jul 2023

Service Reliability Evaluation of Highway Tunnel Based on Digital Image Processing

PONE-D-22-32743R1

Dear Dr. Chunquan,

We’re pleased to inform you that your manuscript has been judged scientifically suitable for publication and will be formally accepted for publication once it meets all outstanding technical requirements.

Kind regards,

Jianguo Wang, PhD

Academic Editor

PLOS ONE

Additional Editor Comments (optional):

Reviewers' comments:

Reviewer's Responses to Questions

**Comments to the Author**

1. If the authors have adequately addressed your comments raised in a previous round of review and you feel that this manuscript is now acceptable for publication, you may indicate that here to bypass the “Comments to the Author” section, enter your conflict of interest statement in the “Confidential to Editor” section, and submit your "Accept" recommendation.

Reviewer #1: All comments have been addressed

2. Is the manuscript technically sound, and do the data support the conclusions?

Reviewer #1: Yes

3. Has the statistical analysis been performed appropriately and rigorously? 

Reviewer #1: Yes

4. Have the authors made all data underlying the findings in their manuscript fully available?

Reviewer #1: Yes

5. Is the manuscript presented in an intelligible fashion and written in standard English?

Reviewer #1: Yes

6. Review Comments to the Author

Reviewer #1: The authors have addressed all the comments raised by the reviewer. The paper may be accepted for publication.

7. PLOS authors have the option to publish the peer review history of their article (what does this mean?). If published, this will include your full peer review and any attached files.

Reviewer #1: No

---

## [Editor Report · Acceptance letter]

1 Aug 2023

PONE-D-22-32743R1 

Service Reliability Evaluation of Highway Tunnel Based on Digital Image Processing 

Dear Dr. Dai:

I'm pleased to inform you that your manuscript has been deemed suitable for publication in PLOS ONE. Congratulations! Your manuscript is now with our production department. 

Kind regards, 

on behalf of

Dr. Jianguo Wang 

Academic Editor

PLOS ONE